# Inclisiran—A Revolutionary Addition to a Cholesterol-Lowering Therapy

**DOI:** 10.3390/ijms24076858

**Published:** 2023-04-06

**Authors:** Adrianna Dec, Aleksandra Niemiec, Eliza Wojciechowska, Mateusz Maligłówka, Łukasz Bułdak, Aleksandra Bołdys, Bogusław Okopień

**Affiliations:** Department of Internal Medicine and Clinical Pharmacology, School of Medicine in Katowice, Medical University of Silesia in Katowice, Medyków 18, 40-752 Katowice, Poland

**Keywords:** hypercholesterolemia, PCSK9 inhibitors, siRNA, inclisiran

## Abstract

Hypercholesterolemia plays a crucial role in the development of atherosclerosis, but it remains an undertreated and underdiagnosed disease. Taking into consideration the high prevalence of lipid disorders, long duration of the asymptomatic course of the disease, life-threatening complications resulting from inaccurate therapy, and stringent treatment goals concerning LDL cholesterol level in the prevention of cardiovascular events, novel lipid-lowering therapies have been introduced in the last few years. In this article, a drug belonging to the group of small interfering RNA (siRNA) called inclisiran is described. It is a novel molecule that increases the number of LDL receptors (LDLRs) on the surface of hepatic cells by preventing the formation of proprotein convertase subtilisin/kexin type 9 (PCSK9) responsible for the degradation of LDLRs. With great potential for lowering plasma LDL cholesterol level, high liver specificity, comfortable dosing regimen, and good tolerance without significant adverse effects, it could play an important part in future hypolipemic therapies.

## 1. Introduction

The definition of dyslipidemia is an abnormal concentration of lipids (cholesterol and/or triglycerides and/or lipoproteins) in the blood [1]. Patients with hyperlipidemia are at a higher risk of developing cardiovascular diseases compared to those with a normal lipid profile. Statins are the first-line treatment method for lowering LDL-C [2]. However, in some cases the use of statins is insufficient, and up to 20% of high-risk patients still cannot reach their LDL-C targets.

In the last several years, new drugs targeting proprotein convertase subtilisin/kexin type 9 (PCSK9) have been introduced to the therapy [3]. Inclisiran is a novel long-acting small interfering RNA (siRNA) that inhibits the synthesis of PCSK9 in the liver and thereby contributes to a reduction in LDL-C levels. The drug was intended to reduce LDL-C in patients with heterozygous familial hypercholesterolemia or established atherosclerotic cardiovascular disease (ASCVD) who are unable to achieve recommended LDL treatment goals with diet and maximally tolerated statin therapy [4]. Inclisiran is associated with improved lipid profile and, according to clinical trials, contributes to a reduction in major adverse cardiac events and hospitalizations for heart failure and strokes when compared to placebo. Additionally, the drug has a favorable safety profile with no serious side effects [5].

The aim of this review is to present inclisiran’s mechanism of action, efficacy, and safety profile in comparison with currently available lipid-lowering drugs.

## 2. Causes of Hyperlipidemia

Dyslipidemia may have a genetic or environmental origin. The most common type of primary lipid disorder is polygenic hypercholesterolemia, which results from an interaction of unidentified, multiple genetic factors accompanied by an unhealthy lifestyle—that is, lack of physical activity and improper diet, which consists of increased intake of carbohydrates (especially monosaccharides) and saturated fat, including trans-fatty acids [6].

Although polygenic causes of dyslipidemia accompanied by unhealthy lifestyle are the main culprit of hyperlipidemia, there is also monogenic disease called familial hypercholesterolemia (FH). FH is an inherited autosomal co-dominant disorder characterized by significantly elevated plasma cholesterol levels, especially low-density lipoproteins (LDL-C), clinically resulting in an increased risk of premature ASCVD [6,7]. Major mutations concerning four genes encoding proteins involved in LDL metabolism have been described in patients with FH: LDL receptor (most common), apolipoprotein B (Apo B), low-density lipoprotein receptor adaptor protein (LDLRAP), and proprotein convertase subtilisin/kexin type 9 (PCSK9). Patients with very high LDL-C levels with no mutations in the aforementioned genes are likely to have a single nucleotide polymorphism or undiscovered genetical defect [8]. The prevalence of familial hypercholesterolemia in its heterozygous form (HeFH) has been estimated to affect 1:300 individuals. The incidence of the homozygous form of the disease (HoFH) ranges between 1 in 160.000 and 1 in 1.000.000 [9].

Hyperlipidemia may also be a secondary condition due to underlying diseases such as uncontrolled diabetes mellitus, chronic kidney disease, Cushing syndrome, polycystic ovarian syndrome, and hypothyroidism [10]. Furthermore, drugs (e.g., glucocorticoids, beta blockers, atypical antipsychotics) and dietary factors—mainly alcohol—may also contribute to the development of dyslipidemia [11].

## 3. The Assessment of Cardiovascular Risk

Nowadays, the management of hypercholesterolemia is focused on the prevention of cardiovascular diseases (CVDs). LDL-C is considered the major cardiovascular risk factor, but in order to assess the individual cardiovascular risk, other factors have to be assessed [12]. Recently, a questionnaire chart, entitled SCORE2 (Systematic Coronary Risk Estimation 2), has been developed to estimate the 10-year risk of cardiovascular incident in Europe [13].

SCORE2 allows to predict the combined 10-year risk of death from cardiovascular causes, as well as the incidence of nonfatal stroke and myocardial infraction. The algorithm is based on well-known risk factors such as age, sex, non-HDL cholesterol level, blood pressure, and cigarette smoking. A new aspect is the classification of four different regions in Europe depending on the risk of cardiovascular incident (low, moderate, high, very high) [13].

To emphasize the pertinence of CVD prevention, it is crucial to remember that according to the World Health Organization, CVDs remain a major cause of morbidity and mortality worldwide [14,15]. The assessment of cardiovascular risk using the SCORE2 chart is recommended for asymptomatic adults (primary prevention) without documented CVD, chronic kidney disease (CKD), diabetes, or familial hypercholesterolemia. The new European Society of Cardiology/European Atherosclerosis Society (ESC/EAS) Guidelines for the management of dyslipidemia present very clear and strict goals that should be achieved while dealing with lipid disturbances. In patients at very high cardiovascular risk (e.g., with a history of myocardial infarction or stroke, concomitant systemic complications of diabetes, or advanced stage of CKD), target LDL-C concentration in the blood is less than 55 mg/dL with simultaneous reduction of at least 50% or more from baseline [16]. Such demanding treatment goals have urged scientists to develop more powerful lipid-lowering drugs, which may facilitate meeting those requirements.

Apart from SCORE charts, there are other algorithms that are commonly used to assess the individual’s cardiovascular risk.

The QRISK chart is more often used in the United Kingdom. It was developed by the National Institute for Health and Care Excellence (NICE). In contrast to SCORE2, its unique feature is the inclusion of social deprivation on the assessment of cardiovascular risk [17].

The Framingham risk score has been validated in the United States [18]. The Framingham Heart Study has given the necessary data to develop the above-mentioned chart to evaluate the 10-year risk of developing coronary artery disease [19].

The PROCAM (Prospective Cardiovascular Münster) system is based on the data obtained from the German PROCAM study. The 10-year risk of coronary events can be estimated using HDL-C, LDL-C, triglycerides, age, family history of CHD, smoking, diabetes occurrence, and systolic blood pressure [20].

## 4. Lipid-Lowering Therapies

### 4.1. Nonpharmacological Methods

The treatment of dyslipidemia is based not only on pharmacological methods but also on lifestyle modifications, including bad dietary habits. One of the main nutritional factors that affects LDL-C concentration is saturated fatty acids (SFAs). It is established that every single percent of energy derived from SFA causes an increase in LDL-C level of 0.8–1.6 mg/dL. On the other hand, oils rich in unsaturated fatty acids such as safflower oil, sunflower oil, rapeseed oil, or linseed oil, when used instead of butter or lard, are able to reduce LDL-C concentration by around 8–16 mg/dL [16,21].

Lifestyle modifications are crucial to improve plasma lipid profiles. The PREDIMED (PREvención con DIeta MEDiterránea) trial has shown that in comparison with a low-fat diet, a Mediterranean diet supplemented with virgin olive or nut oil significantly reduces (by approx. 30%) the frequency of serious CV events [22]. To reduce the concentration of total cholesterol and LDL-C levels, the diet should include high fiber content, phytosterols, and red fermented rice with limited intake of saturated and trans fats. A reduction in triglyceridemia may be achieved by the limitation of carbohydrate and alcohol intake [23].

More attention has recently been paid to phytosterols and functional foods, although there is still not enough evidence to consider them to be significant for the natural course of dyslipidemia and CVDs. It is suggested that enriching the diet with 2 g of phytosterols per day can effectively reduce the concentration of TC and LDL-C by around 7–10%. This is achieved by their competition in the intestinal absorption of cholesterol [24].

Another natural substance that was explored for its lipid-lowering properties is monacolin, derived from fermented rice or red yeast rice (RYR), which is an inhibitor of the 3-hydroxy-3-methyl-glutaryl-coenzyme A (HMG-CoA) reductase, an enzyme that is involved in the synthesis of cholesterol. This mechanism of action is analogous to statins, and 5–7 mg of monacolin is as effective as 20–40 mg of lovastatin [25]. In spite of the same mechanism of action, difficulties with the assessment of the exact dose of monacolin as a dietary supplement as well as the risk of the contamination with mycotoxin (citrinin) found in poorly manufactured RYR have rendered monacolin a pharmacological compound that cannot be substituted for conventional treatment with statins [26,27].

In the management of hypertriglyceridemia, high doses of long-chain fatty acids from the omega-3 group (2–3 g per day) can be helpful. Their systematic intake is able to reduce both postprandial lipemia and TG concentration by around 30% [28].

### 4.2. Statins

Currently, statins remain the drugs of choice in the treatment of lipid disorders. They reduce cholesterol levels by the competitive inhibition of a rate-limiting enzyme in its synthesis in the liver, i.e., HMG-CoA reductase activity. The reduction in intracellular cholesterol availability increases the expression of LDL receptors on hepatocytes’ surface. As a result, LDL-C is taken up to a greater extent, and decreased circulating levels of LDL-C and other lipoproteins rich in ApoB, including TG, are seen [29]. Moreover, statins are known for their pleiotropic features including improved endothelial function, improved stability of atherosclerotic plaques, and anti-inflammatory as well as antithrombotic properties [30].

One of the possible mechanisms mediating the beneficial effects of statins on vascular endothelium is inhibition of Rho and Rho-kinase. This prevents the formation of local adhesion complexes, decreases the sensitivity of vascular smooth muscle cells to calcium in both hypertension and coronary spasm, and increases endothelial nitric oxide synthase (eNOS) expression. Research shows that inhibition of Rho isoprenylation in leukocytes and bones leads to many effects unrelated to cholesterol levels such as mobilization of the endothelial progenitor cells from the bone marrow, stimulation of angiogenesis, and attenuation of the adhesion of leukocytes to the vascular endothelium [31].

Statins are able to reduce an inflammatory process in atherosclerotic plaques and increase plaque stability by inhibition of macrophage activity, reduction in the expression of matrix metalloproteinases (MMPs), and lipid-lowering properties. The inhibition of atherosclerosis progression is also caused by the reduction in the expression of monocyte chemoattractant protein-1, which weakens interactions between monocytes and the endothelium, leading to a decline in monocyte chemotaxis and maturation of macrophages [32].

The first statin introduced to the pharmaceutical market in 1987 was lovastatin. Since that time, six statins, including two semi-synthetic statins (simvastatin and pravastatin) and four synthetic statins (fluvastatin, atorvastatin, rosuvastatin, and pitavastatin), have been marketed, and currently over 200 million people benefit from them [33].

Statins may be divided into lipophilic (simvastatin, fluvastatin, pitavastatin, lovastatin, and atorvastatin) and hydrophilic (rosuvastatin and pravastatin) compounds. The absorption of lipophilic substances is faster, but they must undergo metabolism via CYP450 enzymes. On the other hand, hydrophilic statins are mostly excreted without modification. Previously, it was speculated that an ability of lipophilic statins to enter cells using passive diffusion may improve tissue distribution and lead to greater improvements in cardiovascular outcomes. However, the reason for such an observation may simply be the greater LDL-C-lowering potential of these drugs [34].

Statins are considered to be safe, although some side effects may occur. Myalgia is the most common and benign one, with a documented incidence of around 1–10%. On the other hand, rhabdomyolysis, a life-threatening condition associated with a rise in CK greater than 10×, myoglobinuria, renal impairment, and serum electrolyte abnormalities, is very rare or even incidental. Prior to the introduction of statin therapy, liver function should be assessed, because statins are associated with hepatotoxicity in up to 1–3% of patients. A clinically significant increase in alanine transaminase (ALT) activity is a value exceeding three times above the upper limit of the reference range, which mandates drug withdrawal [35,36]. According to safety data, lipophilic statins seem to be associated with a higher risk of adverse effects, mainly due to muscle damage [34].

The efficacy in lowering of plasma LDL-C levels differs among statins. Older drugs such as lovastatin or simvastatin are able to reduce LDL-C levels by 24 to 41% (lovastatin 10–60 mg/day) and by 47% (simvastatin at maximal recommended dose 80 mg/day) [37,38]. One of the most modern statins and, moreover, the most potent one—rosuvastatin—is capable of reducing LDL-C concentration by 46% to 55% depending on the dose (10 to 40 mg/day) [39]. Similar results were obtained during pharmacotherapy with atorvastatin (80 mg a day), which decreased LDL-C by 50.2% [40].

When the recommended LDL-C level is not achieved using a maximally tolerated dose of the most potent statins (i.e., rosuvastatin or atorvastatin) or severe adverse effects occur, therapy should be supplemented with ezetimibe.

### 4.3. Ezetimibe

Ezetimibe inhibits the cholesterol uptake from food and bile in the gastrointestinal tract, acting at the level of the jejunal brush border by selective inhibition of Niemann-Pick C1-Like (NPC1L1) protein. Despite the impact on the absorption of cholesterol, it does not affect other fat-soluble nutrients [41]. However, due to common polypharmacotherapy of comorbidities, it should be kept in mind that some drug interactions may occur. Research shows that ezetimibe has no influence on cytochrome P450 isozymes. Cholestyramine decreases ezetimibe’s availability, but its simultaneous use with fenofibrate or gemfibrozil increases total ezetimibe concentration in the blood by approximately 1.5 times and may lead to cholelithiasis. Finally, ezetimibe addition to the therapy with warfarin may potentially increase the international normalized ratio (INR) [42]

The most common adverse effects in ezetimibe monotherapy are dyspepsia, diarrhea, and weakness. It has not been associated with increased rates of myopathy or rhabdomyolysis; nevertheless, intensification of muscle pain can be observed after adding ezetimibe to a statin. Although elevation in liver transaminase levels during therapy with ezetimibe has been shown clinically insignificant, it is advisable to check the pretreatment liver enzyme levels [42].

The effectiveness of ezetimibe has been evaluated in many clinical trials. A meta-analysis that included over 2700 subjects has proved that 12 weeks of monotherapy with ezetimibe in a dose of 10 mg per day in hypercholesterolemic patients was associated with a significant 18.5% reduction in LDL-C compared to placebo [43]. Its beneficial impact on cardiovascular risk has been confirmed in the the Improved Reduction of Outcomes: Vytorin Efficacy International Trial (IMPROVE-IT), which demonstrated that ezetimibe, on top of statin treatment, significantly reduced the risk of major cardiovascular events in a group of high-risk patients [44].

### 4.4. Combined Therapy

In response to very stringent LDL-C treatment guidelines, the combination of two drugs should be considered. It allows to achieve the goal of improved efficacy with reduced risk of adverse effects. One of the trials that compared the effectiveness and tolerance of monotherapy with 20 mg rosuvastatin versus a drug combination (ezetimibe 10 mg and rosuvastatin 10 mg) was the RACING (randomized comparison of efficacy and safety of lipid lowering with statin monotherapy versus statin–ezetimibe combination for high-risk cardiovascular disease) trial with 3780 participants with atherosclerotic cardiovascular disease. Results have shown that a smaller dose of a statin combined with ezetimibe was noninferior to higher-intensity statin monotherapy, but combined therapy allowed to obtain a higher percentage of patients with a LDL cholesterol concentration of less than 70 mg/dL (respectively, after 1, 2, and 3 years) in 73%, 75%, and 72% of them in the combination therapy group (ezetimibe + rosuvastatin) and in 55%, 60%, and 58% of participants who were receiving only rosuvastatin in a high dose [45]. It was also confirmed that the combination of ezetimibe with rosuvastatin is capable of reducing LDL-C levels even by 61–65% from baseline (20 mg rosuvastatin; 10 mg ezetimibe) [46].

The efficacy of the combined treatment strategy has also been demonstrated in patients with homozygous familial hypercholesterolemia. A randomized, double-blind, parallel-group study has shown that ezetimibe with statin in a dose of 80 milligrams of atorvastatin or simvastatin significantly reduced LDL-C levels compared to statin monotherapy (−27.5% versus −7%) [47]. Davidson and colleagues in a study of 2382 patients have demonstrated that the maximal LDL-C drop while using combined therapy was observed 2 weeks after the initiation of treatment with ezetimibe. Additionally, the safety and supreme potency to decrease LDL-C concentration using coadministration of 10 milligrams of ezetimibe with a statin compared to the statin alone was noticed [48].

Further safety analyses of combined therapy have also not indicated a greater propensity to adverse effects compared to statin monotherapy or placebo [49].

### 4.5. PCSK9 Inhibitors

If the aforementioned drugs are insufficiently effective and therapeutic goals are not achieved, then further therapeutic steps should be undertaken. The following step in hypolipemic therapy should target proprotein convertase subtilisin/kexin type 9 (PCSK-9).

#### 4.5.1. PCSK9

Plasma LDL-C concentration depends greatly on its removal from the blood by the liver. The effectiveness of this process is related to the expression of LDL receptors (LDLRs) on the hepatocytes’ surface. When LDL attaches to a dedicated receptor, then the formed complex is internalized by clathrin-mediated endocytosis and directed to endosomes. There, LDL detaches from LDLR and is further transported to the lysosomes, where it undergoes degradation. In turn, the LDLR is transported back to the cell surface and is able to attach more LDL particles from systemic circulation [50].

In 2001 in cultured cerebellar granule neurons, scientists identified the presence of neuronal apoptosis regulated convertase-1 (NARC-1). It has also been shown that in these cells NARC-1 mRNA is upregulated in response to exposure to pro-apoptotic signals. The above-described protein playing a role in the mechanism of apoptosis required for the development of the nervous system is currently known under the name of proprotein convertase subtilisin/kexin 9 (PCSK9) [51,52].

PCSK9 is one of the most important regulators of the number of LDLRs. It belongs to one of nine known proprotein convertases (PCs) that serve as serine proteases involved in post-translational modifications of propeptides. *PCSK9* is expressed mainly in the liver, but its presence in smaller amounts has also been confirmed in the intestines, kidneys, and brain [53]. Its gene is located on chromosome 1p32.3, and the protein is synthesized as a 74 kDa zymogen in the endoplasmic reticulum [53,54].

PCSK9 consists of a signal peptide, a prodomain, a catalytic domain, and a C-terminal domain [50]. It undergoes many post-translational modifications. One of them is autocatalytic activation by cleavage at the N-terminus, which leads to the formation of mature protein destined for release into the bloodstream [55]

Some of the mature PCSK9 particles undergo additional cleavage by another proprotein convertase called furin [56]. Significant differences between the two mentioned forms of PCSK9 have been shown. It is considered that the mature forms of PCSK9, unlike the furin-cleaved forms, are responsible for regulating the amount of LDLR. Therefore, mature forms seem to relate to atherogenic properties [57,58,59]. Furthermore, *PCSK9* loss-of-function point mutations that impede recognition and proteolysis by furin enhance LDLR degradation [55]. This regularity has been noted in the group of patients with FH and a mutation of *PCSK9* reducing its susceptibility to furin. Increased secretion of the uncleaved protein was associated with higher concentration of LDL-C [58].

After releasing to plasma, PCSK9 binds to epidermal growth factor homology domain A (EGF-A) of the LDL receptor. That leads to endocytosis of a newly formed complex and digestion of LDLR by hepatocyte lysosomes. By reducing the number of LDL receptors on the hepatocyte surface, PCSK9 increases LDL-C levels [60,61]. Some studies have shown that PCSK9 can also lead to LDLR degradation by intracellular pathway, before its exocytosis via the Golgi–lysosome route [50,62,63].

A breakthrough in understanding the role of PCSK9 was the discovery made by Abifadel and colleagues in 2003, who identified two missense mutations in *PCSK9* genes (S127R, F216L) as the cause of autosomal dominant familial hypercholesterolemia [64]. In subsequent years, studies have shown that similar mutations increasing *PCSK9* activity (gain-of-function mutations) are associated with increased LDL-C levels, while loss-of-function mutations result in decreased LDL-C levels and decreased risk of CVD [62].

The discovery of PCSK9 impact on lipid metabolism paved the way for a novel class of lipid-lowering agents involving PCSK9 inhibition:Monoclonal antibodies (mAbs)—alirocumab and evolocumab;Small interfering RNA (siRNA)—inclisiran.

Unlike mAbs, which block only the extracellular mechanism of PCSK9 action, inclisiran acts on both intra- and extracellular pathways [50].

#### 4.5.2. Anti-PCSK9 Monoclonal Antibodies

Currently, the only registered anti-PCSK9 mAbs are two fully human monoclonal antibodies: alirocumab and evolocumab. The best results of therapy are obtained when they are used in combination with statins, because statin therapy increases circulating PCSK9 levels in serum [65].

In the FOURIER study, evolocumab or placebo was administered every 2 or 4 weeks to 27.564 patients with ASCVD, LDL-C > 70 mg/dL, or non-HDL-C > 100 mg/dL who were taking an optimal dosage of statin. The primary combined end point (CV death, myocardial infarction, stroke, hospitalization for unstable angina, or coronary revascularization) occurred at a significantly lower rate at 2.2 years of follow-up in the evolocumab group (1344 patients, 9.8%) compared to the placebo group (1563 patients, 11.3%, *p* = 0.001). Apart from a minor increase in incidence of injection site reactions in the evolocumab group, there were no significant differences in adverse events between both groups [66,67].

The ODYSSEY study showed that alirocumab was safe and effective for patients who had experienced an ACS incident in the year prior and were on maximally tolerated statin treatment. A total of 18.924 individuals were recruited and given either alirocumab 75–150 mg every 2 weeks (to maintain LDL-C between 25 and 50 mg/dL) or placebo following an initial run-in period of 2–16 weeks on high-intensity statin treatment. MACEs were significantly reduced in alirocumab-treated participants compared to the placebo group (9.5% vs. 11.1%, *p* = 0.001). [67]. Furthermore, the CHOICE II trial has demonstrated alirocumab’s efficacy in statin-intolerant patients, which translated into a 56% reduction in LDL (vs. placebo; *p* < 0.001) [68].

The results of the therapy with evolocumab were similar. In the GAUSS-2 trial with 307 statin intolerant patients, treatment with evolocumab reduced LDL-C levels by 56% vs. 39% in other groups (placebo + ezetimibe) [69].

Anti-PCSK9 mAbs also have been evaluated in patients with familial hypercholesterolemia. The ODYSSEY HIGH FH trial reported 105 patients with familial hypercholesterolemia using maximally tolerated statin therapy and LDL >160 mg/dL. In this study, alirocumab obtained a 46% reduction in LDL (vs. 7% with placebo) [30]. Evolocumab was also successfully used in patients with heterozygous and homozygous familial hypercholesterolemia. The RUTHERFORD-2 trial, which enrolled 329 patients with FH, has shown a significant reduction in LDL with both evolocumab regimens: 140 mg every 2 weeks led to a 59.2% reduction and 420 mg once a month led to a LDL reduction of 61.3% compared to placebo [70].

## 5. Inclisiran—A Novel PCSK9 Inhibitor

Despite the widely available effective lipid-lowering drugs, treatment goals are not always achieved, which may result from a patient’s reluctance to take statins, low adherence, or therapeutic inertia. A breakthrough approach to this problem may be an inclisiran—a novel hypolipemic drug that impedes the production of PCSK9.

### 5.1. siRNA

In 1998, American scientists Andrew Fire and Craig Mello described a phenomenon of mRNA degradation resulting in gene expression silencing. For explaining the mechanism of RNA interference (iRNA), in 2006 both received the Nobel Prize in Physiology or Medicine [60].

Post-transcriptional gene silencing—a process that occurs in the majority of eukaryotic cells, is the basis of siRNA action. siRNA consists of 21–23 nucleotides and arises as a result of cleaving of dsRNA by a specific ribonuclease called Dicer (schematic structure shown in Figure 1). It contains a passenger strand and a guide strand, both with a phosphorylated 5′-end and a hydroxylated 3′-end [71,72]. siRNA is recognized and attached to a RNA-induced silencing complex called RISC. Within the formed complex, an interaction with the Argonaute 2 protein takes place, which results in the unwinding and degradation of the passenger strand. Thermodynamic stability in base pairing at the 5′-end influences the selection of which strand becomes leading—the strand with the less stable base paired on the 5′-end is recognized by RISC as a guide strand. The guide strand loaded on the RISC allows for binding with the target mRNA and then consequently undergoes its cleaving and the process of translation is blocked, which results in a decreased level of the resultant protein [60,71,73].

The discovery of iRNA has paved the way for new drug therapies, but many obstacles stand in the way of achieving its intended effects. The first challenge was finding the right way to administer the drug—oral administration led to rapid degradation and unsatisfactory absorption. This issue was resolved by intravenous and subcutaneous administration or direct application to the target tissue (e.g., eye, skin) [73]. Efficacy and safety of siRNA is achieved by several modifications including the addition of moieties, creating conjugates or polymers [70]. SiRNA technology provides several significant advantages compared to conventional, well-known forms of pharmacotherapy. First of all, the potential inhibitory effect on the expression of an unlimited number of genes. Currently, the search for the targeted application methods in various diseases should be emphasized. High selectivity and reversibility (siRNA does not lead to permanent modification of the genome) should vouch for its safety. Compared to other novel pharmaceuticals such as monoclonal antibodies, it might become less expensive in production and become smaller burden for the healthcare budget [73,74].

At the moment, there are four iRNA-based drugs approved by the US Food and Drug Administration and used in clinical practice:Patisiran (2018) for hereditary transthyretin-mediated amyloidosis treatment;Givosiran (2019) for acute hepatic porphyria therapy;Lumasiran (2020) for the treatment of primary hyperoxaluria type 1;Inclisiran (2021) as a hypolipidemic drug.

Furthermore, several siRNAs (e.g., olpasiran) are subjects of ongoing studies. Therefore, one might expect the oncoming years to bring new discoveries and therapeutic opportunities [75].

### 5.2. Inclisiran

#### 5.2.1. Mechanism of Action

Inclisiran, a novel synthetic siRNA, exploits the natural mechanism of protein synthesis inhibition. It binds to the mRNA of the PCSK9 precursor and causes its degradation. This results in reduced PCSK9 protein production and, consequently, increased presence of LDLR, which improves the efficiency of plasma LDL-C clearance and decreases its level in circulation.

Inclisiran molecules are linked to the ligand triantenary N-acetylgalactosamine (GalNAc), which has an affinity for the asialoglycoprotein receptors found mainly on hepatocytes [60,75,76]. GalNAc-siRNA conjugates bind to ASGPRs, which causes rapid receptor-mediated endocytosis of the formed complex. The separation of GalNAc-siRNA from ASGPR is results from the decrease in endosomal pH. The released asialoglycoprotein receptor returns to the cell surface and can attach further GalNAc-siRNA conjugates [73,77]. Inclisiran, which is gradually released to cytoplasm from endosomes, consists of two strands: the passenger strand—needed for loading the drug molecule to the RNA-induced silencing complex (RISC)—and the guide strand, which carries information required for target mRNA recognition. The guide strand loaded on the RISC hybridizes to the *PCSK9* mRNA and causes its cleavage [62,77]. The mechanism of action of inclisiran is schematically shown in Figure 2.

The presence of the GalNAc ligand makes the action of inclisiran highly selective. While monoclonal antibodies bind to circulating PCSK9, the main target for inclisiran is PCSK9 production in hepatocytes [62]. This, in turn, reduces the risk of side effects from other organs and also allows a lower cumulative dose to be used [62]. However, coupling with GalNAc is not the only modification that inclisiran has undergone. The use of 2′-fluoro and 2′-O-methyl modified nucleotides increases the stability of the drug. The addition of phosphorothioate ligands also performs a similar function by preventing siRNA from degradation by exonucleases [78].

Several additional factors are responsible for the long-term effect of inclisiran. For example, the RISC protein complex formed upon inclisiran attachment can cleave multiple copies of *PCSK9* mRNA [60,72]. It is assumed that inclisiran-loaded RISC has a half-life of several weeks [60]. Furthermore, only a small number of inclisiran molecules remain biologically active in a certain period of time. The majority, after reaching the hepatocytes, are stored in endosomes, from which they are gradually released [60].

Inclisiran is administered subcutaneously in a single injection at a dose of 300 mg. It should be administered as initial dose, then after 3 months, and then consecutively at 6-month intervals (whereas PCSK9-inhibiting monoclonal antibodies require injections every 2–4 weeks). The target site of injection is the abdomen, but alternatively inclisiran can be also administered in the upper arm or thigh, excluding areas where the skin is damaged or irritated [79]. Approximately 4 h after administration, it reaches its maximum plasma concentration. The half-life of inclisiran is 9 h; after 24–48 h post-dosing, it reaches an undetectable level in the circulation. Nonspecific nucleases are responsible for its metabolism. It is assumed that it is not a substrate for CYP450 or any other drug transporter. Approximately 16% of inclisiran is eliminated through the kidney [10,64,79].

#### 5.2.2. Safety and Adverse Effects

Inclisiran seems to be a solution for people with statin treatment intolerance. ORION trials results showed that it lowers LDL-C levels with a good safety profile. Most of the reported adverse effects of inclisiran were mild and related to injection site reactions such as pain or rash. Other off-target effects observed during its use included systemic response resembling infectious symptoms e.g., fever, musculoskeletal pain, sore throat, headache, fatigue, or nasopharyngitis [72,80,81]. Furthermore, previous trials have shown a similar frequency of side effects in the group of people using placebo and inclisiran—in the ORION-1 study, 76% in both groups. However, in the group of patients receiving the drug, serious side effects were observed slightly more often—in 11%, compared to 8% in the placebo group [61]. In the cited study, one patient experienced an asymptomatic increase in liver enzymes level (γ-glutamyltransferase and alanine aminotransferase), but it was assumed that the adverse event was related to the concomitant use of a statin rather than inclisiran itself [61,72].

In the ORION-3 study, a 4-year open-label extension study of ORION-1, side effects were observed in 275 of 284 (97%) patients in the inclisiran-only arm and in 80 of 87 (92%) patients in the evolocumab switched to inclisiran arm—mostly mild and self-limiting. In 79 of 284 (28%) in the first group and in 22 of 87 (25%) in the second one, side effects were deemed as related to the examined drug. There was a difference in the spectrum of the most common side effects between the groups. The patients in the inclisiran-only arm experienced nasopharyngitis (55 of 284—19%), while those in the switching arm suffered from hypertension (17 of 87—20%). Serious adverse events occurred in 104 of 284 (37%) of the participants in the inclisiran arm and in 30 of 87 (34%) in the switching arm. When it comes to impaired liver function, there were 10% (28 of 284) of such cases in the inclisiran-only group and 9% (8 of 87) in the switching arm. There were eight deaths during the ORION-3 study—seven reported in the inclisiran-only group and one in the switching group. None of fatal outcomes seemed to be a direct effect of the drug [82].

The safety of inclisiran may be related to its high specificity for hepatocytes. Inclisiran molecules are conjugated to N-acetylgalactosamine carbohydrates (GalNAc), complementary to hepatocyte asialoglycoprotein receptors, which results in rapid uptake only into the liver [60,61]. Selective action reduces the risk of peripheral neuropathies reported during other siRNA therapies [61,83]. Phosphorothioate modifications of inclisiran (2′—O—methyl nucleotide and 2′—fluoro nucleotide) increase its stability and reduces the risk of immunogenicity [62]. Nevertheless, these modifications may trigger platelet-activating factor and potentially increase the risk of thrombosis. However, this effect has not been observed in previous studies, nor has the activation of the immune response (cellular response and the production of proinflammatory cytokines such as IL-6 or TNF-α were not affected) [61,80,84].

The effect of inclisiran on immunogenicity was studied in a group of 1800 patients. The presence of anti-drug antibodies was detected in 1.8% of participants before dosing and in 4.9% during 18 months of drug administration. However, the presence of antibodies has not affected drug efficacy [79]. Comparably, in the group of 4747 patients using alirocumab, the presence of anti-drug antibodies was found in 5.1% compared to 1% in the placebo group [85].

It is recommended to avoid the use of inclisiran in the group of pregnant women or women trying to become pregnant mainly due to residual content in the liver, which is detectable up to 2 years after one dose [10]. In animal studies, no harmful effects on the well-being of the fetuses have been noted, but its use in pregnant women requires further studies. To date, there is no information on the secretion of the drug into human milk, but animal tests have shown this process to occur [10,79].

Inclisiran can be used in patients with impaired renal function and does not require dose adjustment [76]. On the basis of the ORION-1 and ORION-7 studies analysis, the pharmacodynamics of the drug were comparable in patients with normal and impaired renal function. Increased plasma concentrations were noted and correlated positively with the degree of renal dysfunction, but the drug was not detectable in plasma in either group at 48 h [86,87].

To date, in contrast to statin therapy, there are no indications that pharmacological inhibition of PCSK9 increases the incidence of diabetes [88].

In patients with hepatic impairment (Child-Pugh class A and B), inclisiran seems to be safe in standard doses. However, it has not been tested in patients with severe liver damage (Child-Pugh class C). Therefore, it should be used with caution in this population [79].

Because inclisiran does not inhibit or increase cytochrome P450 activity, it is expected that it should not have clinically significant interactions with other drugs, including statins [79]. The main differences between anti-PCSK9 monoclonal antibodies and inclisiran are presented in Table 1.

#### 5.2.3. Efficacy

Three Phase III placebo-controlled lipid-lowering trials, comprising patients at high risk of cardiovascular events, have revealed that inclisiran reduces circulating PCSK9 and LDL-C levels. However, the hypothesis that lowering LDL-C with inclisiran simultaneously reduces the risk of CV events has not yet been confirmed in randomized controlled trials. It will be evaluated in ORION-4 (NCT03705234) and VICTORION—2 Prevent (NCT05030428) studies [92].

Pooled, patient-level analysis of ORION-9, -10, and -11 included patients with heterozygous familial hypercholesterolemia (HeFH), atherosclerotic CV disease (ASCVD), or risk equivalent of ASCVD on pharmacotherapy with maximally tolerated doses of statin. Participants were randomized 1:1 to receive 284 mg inclisiran or placebo on Days 1, 90, and then every 6 months thereafter for 18 months. [93] Mean LDL-C level at baseline was 112 mg/dL. At Day 90, the absolute reduction in LDL-C was 53 mg/dL. In comparison to the placebo group, percentage reduction in LDL-C with inclisiran reached 50.6%. Prespecified endpoint of major cardiovascular events (MACEs) included cardiac arrest, unclassified death due to cardiovascular reason, fatal and nonfatal myocardial infarction, and fatal and nonfatal stroke. Among 3655 high-risk patients enrolled in the trials, 303 of them were reported with MACEs, including 28 patients with fatal and nonfatal stroke and 74 patients with fatal and nonfatal myocardial infarction. Presented data have shown that inclisiran added to background lipid-lowering therapies was associated with a 26% reduction in MACEs (*p* < 0.05) [94]. Figure 3 presents the dosing schedule and assessment time points of the ORION-9, -10, and -11 clinical trials.

The safety and efficacy of inclisiran were also assessed in patients with homozygous familial hypercholesterolemia (HoFH) in the ORION-5 study [95,96]. The disease, as a consequence of mutations in *LDLR*, *APOB*, or *PCSK9* genes inherited from both parents, is characterized by extremely elevated LDL-C levels and, subsequently, high risk for premature cardiovascular diseases [97,98]. Participants who met the inclusion criteria were randomly assigned to receive either a dose of 300 milligrams of inclisiran or a placebo. An absolute change in LDL-C level from baseline up to day 180 in inclisiran group was (−14.5) mg/dL [(−47.5) − 18.5] versus (−7.0) mg/dL [(−50.5) −36.4] in the placebo group—*p*-value = 0.7861 [96]. To date, statins are still the first-choice treatment with an estimated LDL-C reduction between 6% and 28%. Combined therapy with statin and ezetimibe may cause further lowering of LDL-C concentration by 20.5% [99]. In this group of patients, conventional therapies are generally not able to reduce LDL-C to even satisfactory levels, and inclisiran might be a new, promising therapeutic option.

The current approach to lipid-lowering therapies focuses not only on LDL-C reduction. Novel hypolipemic compounds are currently investigated for other types of lipid molecules responsible for increased CVD risk, e.g., lipoprotein a [lp(a)]—non-LDL-C apoB-containing particle [100]. It has been established that inclisiran is able to reduce lp(a) concentration in approximately 17–25% from baseline [96]. Statins and ezetimibe have shown no substantial effect on lp(a) level, whereas anti-PCSK9 monoclonal antibodies can reduce it by 25–30% [101,102]. These data show a potential benefit in reducing the residual cardiovascular risk by therapies focusing on PCSK9.

As a result of the above-mentioned data, inclisiran (in addition to a low-fat diet and maximally tolerated dose of statin) has been approved by the European Medicines Agency (EMA) for lowering cholesterol levels in patients with primary hypercholesterolemia or mixed dyslipidemia and by the United States Food and Drug Administration for the therapy of heterozygous familial hypercholesterolemia (HeFH) and clinically evident atherosclerotic cardiovascular disease [94,95].

## 6. Possible Future Lipid-Lowering Treatment

The promising effects of PCSK9 inhibitors have inspired researchers to look for alternative ways to affect the PCSK9-dependent hypolipidemic pathway. An interesting approach involves CRISPR/Cas9—clustered regularly interspaced short palindromic repeats/CRISPR-associated protein 9—a mechanism based on genome editing that has been observed for the first time in bacteria as a method of adaptive immunity against viral infections. Fragments of viral DNA derived from previous infections are employed to fight similar bacteriophages attack in the future [103].

CRISPR requires two elements—a guide RNA and Cas proteins. A guide RNA is necessary for sequence-specific targeting, while Cas proteins play structural role. The RNA binds to a Cas protein and cuts the target genome. Cas9 protein and a guide RNA bind together and contribute to efficient cleavage and deletions of specific loci, which could render protein-coding genes inactive [104].

The technique of genome editing has been intensively expanded in biomedical research. The usefulness of genome editing techniques has been investigated, inter alia, in the field of cardiovascular disorders [105,106]. Genome-sequencing data give an opportunity to understand how genetic variations affect lipid disorders. Recent experimental studies suggest that CRISPR gene-editing that targets *PCSK9* might be a promising tool to achieve the ultimate resolution of elevated LDL-C due to a lifelong therapeutic effect with just a single dose of the drug [103].

According to a recent study, the CRISPR/Cas9 method can effectively and permanently damage *PCSK9* expression when used in vivo [107]. Ding et al. treated mice with an adenovirus expressing *S. pyogenes* Cas9 and a guide RNA directed at exon 1 of *PCSK9*. Significant nonhomologous end joining (NHEJ)-mediated mutagenesis was confirmed 3–4 days after injection into mice livers. More importantly, this was accompanied by a substantial drop in blood PCSK9 protein levels by 90% and a concurrent 35–40% reduction in total blood cholesterol levels [108].

Similarly, positive effects were obtained in a study using CRISPR/Cas9 in humanized-liver mouse models (Wang et al. [108]), with a more than 50% reduction in PCSK9 levels [109,110]. Unlike the previous two researchers, Ran and colleagues used Cas9 derived from *S. aureus*. It was small enough to be delivered by adeno-associated viral vectors and resulted in a decrease in PCSK9 concentration of >90% and cholesterol of around 40% [108,110]. VT—1001 is an open-label phase 1 study to assess the safety of VERVE-101 (CRISPR technology) in a single dose in humans [89].

Apart from the CRISPR strategy, other next-generation PCSK9 inhibitors remain under development:

-Lerodalcibep (LIB003): extracellular PCSK9-binding fusion protein [110,111];-AZD8233 (ION449): antisense oligonucleotide conjugated with GalNAc for intrahepatic PCSK9 degradation [110,112];-CIVI-008: capadecursen sodium, third-generation antisense oligonucleotide for *PCSK9*’s expression inhibition [110];-MK-0616: oral synthetic cyclic peptide inhibiting PCSK9 for daily administration [110,113];-NNC0385-0434: peptide inhibitor of PCSK9 used orally [110];-CVI-LM001: first-in-class oral small molecule PCSK9 modulator inhibiting *PCSK9* transcription and degradation of LDLR mRNA [110,114];-PCSK9 vaccine (AT04A, AT06A): mimicking the domain of PCSK9 on its N-terminus in phase I study [111,112,113,114,115,116]

## 7. Conclusions

Inclisiran is a novel lipid-lowering drug that interferes with the PCSK9-dependent hypolipidemic pathway, and it can play a major role in the therapy of dyslipidemia in the future. Designed primarily for patients at high or very high cardiovascular risk, to date it has presented very promising results in lowering LDL-C levels. Current data on its safety profile, lack of serious side effects, and convenient dosing regimen substantiate the high clinical usefulness of this drug. The results of ongoing double-blinded randomized controlled trials assessing the reduction in MACE incidence and improving ASCVD outcomes are warranted to establish its role in the current pharmacotherapy of lipid disorders.

## Figures and Tables

**Figure 1 ijms-24-06858-f001:**
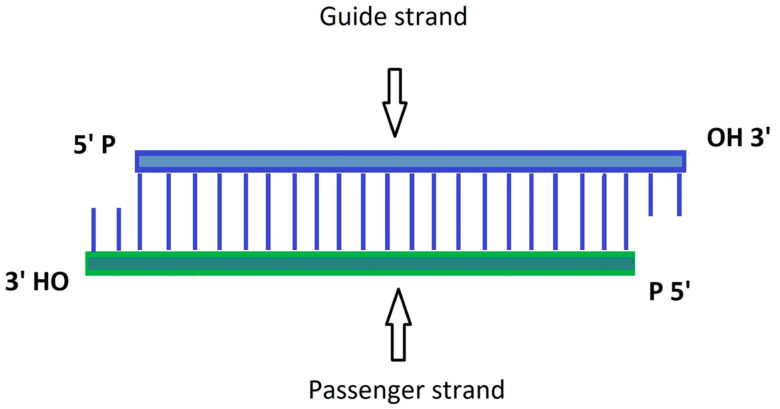
Schematic representation of siRNA structure.

**Figure 2 ijms-24-06858-f002:**
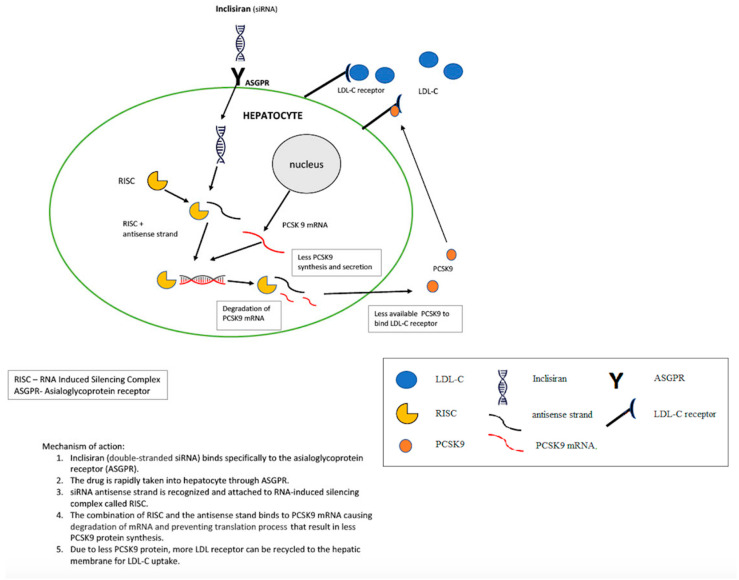
Mechanism of action of inclisiran.

**Figure 3 ijms-24-06858-f003:**
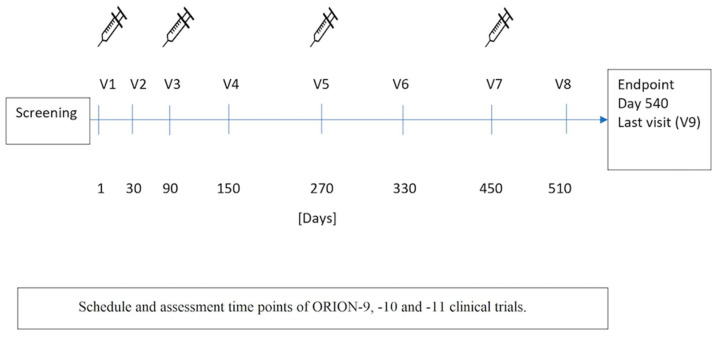
Dosing schedule and time points assessment in ORION-9, -10 and -11 clinical trials.

**Table 1 ijms-24-06858-t001:** Comparison of anti-PCSK9 monoclonal antibodies and inclisiran [82,88,89,90,91].

	Anti-PCSK9 Monoclonal Antibodies (Evolocumab, Alirocumab)	siRNA Targeting PCSK9 (Inclisiran)
Route of administration	Injectable (s.c.)	Injectable (s.c.)
Dose	140 mg/420 mg	284 mg
Dosing frequency	Every two weeks or once monthly	0–90–180 days and every 6 months thereafter
LDL-C plasma level reduction [88,89]	~ 50–60%	~ 50%
PCSK9 reduction [82,89]	~90%	~ 60–80%
Relative reduction in cardiovascular events	~15%	~17%
Mechanism of action	Blocking of the extracellular interaction of PCSK9 LDLR	PCSK9 synthesis inhibition through RNA interference
Advantages	- High specificity	- No serious adverse events - High specificity - Long-term effect - Infrequent dosing
Disadvantages	- Frequent subcutaneous dosing - Short expiry date - High cost	- High cost
Adverse effects [90,91]	- Injection site reaction - Bronchitis (alirocumab) - Diabetes (evolucumab) - Pancreatitis	- Injection site reaction - Bronchitis

## Data Availability

No new data were created or analyzed in this study. Data sharing is not applicable to this article.

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
