# Peer review of "Inclisiran—A Revolutionary Addition to a Cholesterol-Lowering Therapy"

_ijms, 2023, doi:10.3390/ijms24076858_

Round 1
Reviewer 1 Report
In this review, Dec and colleagues present the main lipid lowering therapies, including Inclisiran, a siRNA targeting PCSK9, capable to increase the number of LDLRs and consequently to decrease plasma LDL cholesterol levels. The review should be completed by adding few paragraphs or expanding others (i.e combined therapy, anti-PCSK9 mAb) which were only briefly discussed.
More in specific:
In the chapter of statins more emphasis should be done on the pleiotropic effects of statins including the definition of pleiotropic as cholesterol-independent effects. A brief description of the mechanisms behind these pleiotropic effects (i.e. inhibition of isoprenoids synthesis) is recommended as well as the important distinction between lipophilic and hydrophilic statins and the influence of these properties on the side effects of statins. For example, myalgia is particularly associated with the use of a lipophilic statins.
In the chapter of combined therapy the authors mentioned only two studies both related to combination of rosuvastatin and ezetimibe, while the literature is much more wider. Few examples that can be included and discussed:
Gagne C, Bays HE, Weiss SR, et al. Efficacy and safety of ezetimibe added to ongoing statin therapy for treatment of patients with primary hypercholesterolemia. Am J Cardiol 2002; 90:1084–1091.
Davidson MH, Ballantyne CM, Kerzner B, et al. Efficacy and safety of ezetimibe coadministered with statins: randomised, placebo-controlled, blinded experience in 2382 patients with primary hypercholesterolemia. Int J Clin Pract 2004; 58:746–755.
Kerzner B, Corbelli J, Sharp S, et al. Efficacy and safety of ezetimibe coadministered with lovastatin in primary hypercholesterolemia. Am J Cardiol 2003; 91:418–424.
In the chapter of PCSK9 inhibitors the author only briefly explained the different forms of PCSK9 and their importance for lipids metabolism. In particular, the plasma presence of a furin-cleaved form and its importance for the regulation of turnover of LDLR should be mentioned together with the appropriate literature.
The chapter of anti-PCSK9 monoclonal antibodies is incomplete, missing several references and informations. There are clinical trials not mentioned. Please refer for example to the following similar paper, reporting several studies on the use of PCSK9 Abs:
PMID: 36839781
In table 1 all the infos related to LDL-C plasma reduction, PCSK9 reduction, should have the appropriate reference.
Before the conclusion the author could add a paragraph of Future perspectives where they can mention the very promising approach of gene editing using CRISPR/CAS9 technologies as an additional potential therapy against hypercholesterolemia.
Author Response
Dear Reviewer 1,
I would like to thank you for careful revision of manuscript entitled “Inclisiran – a revolutionary addition to a cholesterol lowering therapy” and relevant remarks that improve the substantive rank of this work. The replies to your comments are presented as changes in paragraphs with the use of red font.
Point 1:
In the chapter of statins more emphasis should be done on the pleiotropic effects of statins including the definition of pleiotropic as cholesterol-independent effects. A brief description of the mechanisms behind these pleiotropic effects (i.e. inhibition of isoprenoids synthesis) is recommended as well as the important distinction between lipophilic and hydrophilic statins and the influence of these properties on the side effects of statins. For example, myalgia is particularly associated with the use of a lipophilic statins.
Response 1:
Authors have implemented additional data on statins according to Reviewer’s suggestion:
“4.2 Statins
Currently, statins remain the drugs of choice in the treatment of lipid disorders. They reduce cholesterol level by the competitive inhibition of a rate-limiting enzyme in its synthesis in the liver, i.e., HMG-CoA reductase activity. The reduction of intracellular cholesterol availability increases the expression of LDL receptors on hepatocytes' surface. As a result, LDL-C are taken up to a greater extent and decreased circulating levels of LDL-C and other lipoproteins rich in ApoB, including TG are seen [29]. Moreover, statins are known for their pleiotropic features including improved endothelial function, improved stability of atherosclerotic plaques and anti-inflammatory as well as antithrombotic properties [30].
One of the possible mechanisms mediating beneficial effects of statins on vascular endothelium is inhibition of Rho and Rho-kinase. It prevents the formation of local adhesion complexes, decreases the sensitivity of vascular smooth muscle cells to calcium in both hypertension and coronary spasm, and increases endothelial nitric oxide synthase (eNOS) expression. Research shows that inhibition of Rho isoprenylation in leukocytes and bones leads to many effects unrelated to cholesterol levels such as mobilization of the endothelial progenitor cells from the bone marrow, stimulation of angiogenesis and attenuation of the adhesion of leukocytes to the vascular endothelium [31].
Statins are able to reduce an inflammatory process in atherosclerotic plaques and increase plaque stability by inhibition of macrophages activity, reduction in the expression of matrix metalloproteinases (MMPs) and lipid lowering properties. The inhibition of the atherosclerosis progression is also caused by the reduction of the expression of monocyte chemoattractant protein-1, which weakens interactions between monocytes and the endothelium, leading to a decline in monocyte chemotaxis and maturation of macrophages [32].
The first statin introduced to the pharmaceutical market in 1987 was lovastatin. Since that time 6 statins, including 2 semi-synthetic statins (simvastatin and pravastatin) and 4 synthetic statins (fluvastatin, atorvastatin, rosuvastatin and pitavastatin) have been marketed and currently over 200 million people benefit from their precious action [33].
Statins may be divided into lipophilic (simvastatin, fluvastatin, pitavastatin, lovastatin and atorvastatin) and hydrophilic (rosuvastatin and pravastatin) compounds. The absorption of lipophilic substances is faster, but they must undergo metabolism via CYP450 enzymes. On the other hand, hydrophilic statins are mostly excreted without modification. Previously it was speculated that an ability of lipophilic statins to enter cells using passive diffusion may improve tissue distribution and lead to greater improvements in cardiovascular outcomes. However, the reason for such observation may simply be a resultant of a greater LDL-C lowering potential of those drugs [34].
Statins are considered to be safe, although some side effects may occur. Myalgia is the most common and benign one, with documented incidence around 1-10%. On the other hand, rhabdomyolysis which is a life-threatening condition associated with a rise in CK greater than 10x, myoglobinuria, renal impairment and serum electrolyte abnormalities is very rare or even incidental. Prior to the introduction of statin therapy, liver function should be assessed, due to the fact that statins are associated with hepatotoxicity in up to 1-3% of patients. Clinically significant increase in alanine transaminase (ALT) activity is a value exceeding 3 times above the upper limit of reference range, which mandates drug withdrawal [35, 36]. According to safety data, lipophilic statins seem to be associated with a higher risk of adverse effects, mainly due to muscle damage [34]
The efficacy in lowering of plasma LDL-C level differs among statins. Older drugs like lovastatin or simvastatin are able to reduce LDL-C level by 24 to 41% (lovastatin 10–60 mg/day) and by 47%, (simvastatin at maximal recommended dose 80mg/day) [37, 38]. One of the most modern statins and moreover the most potent one - rosuvastatin is capable of reducing LDL-C concentration by 46% to 55% depending on the dose (10 to 40 mg/day) [39]. Similar results were obtained during pharmacotherapy with atorvastatin (80 mg a day), which decreased LDL-C by 50.2% [40].
When recommended LDL-C level is not achieved using maximally tolerated dose of the most potent statins (i.e., rosuvastatin or atorvastatin) or severe adverse effects occur, therapy should be supplemented with ezetimibe.”
[31] Liao, J.K.; Laufs, U. Pleiotropic effects of statins. Annu Rev Pharmacol Toxicol. 2005;45:89-118.
[32] Kavalipati, N.; Shah, J.; Ramakrishan, A.; Vasnawala, H. Pleiotropic effects of statins. Indian J Endocrinol Metab. 2015, 19(5):554-62.
[34] Climent, E.; Benaiges, D.; Pedro-Botet J. Hydrophilic or Lipophilic Statins? Front Cardiovasc Med. 2021, 8:687585.
Point 2:
In the chapter of combined therapy the authors mentioned only two studies both related to combination of rosuvastatin and ezetimibe, while the literature is much more wider. Few examples that can be included and discussed:
Gagne C, Bays HE, Weiss SR, et al. Efficacy and safety of ezetimibe added to ongoing statin therapy for treatment of patients with primary hypercholesterolemia. Am J Cardiol 2002; 90:1084–1091.
Davidson MH, Ballantyne CM, Kerzner B, et al. Efficacy and safety of ezetimibe coadministered with statins: randomised, placebo-controlled, blinded experience in 2382 patients with primary hypercholesterolemia. Int J Clin Pract 2004; 58:746–755.
Kerzner B, Corbelli J, Sharp S, et al. Efficacy and safety of ezetimibe coadministered with lovastatin in primary hypercholesterolemia. Am J Cardiol 2003; 91:418–424.
Response 2
Authors have edited the paragraph concerned combined hypolipemic therapy and implemented proposed references:
“4.4 Combined therapy
In response to very stringent LDL-C treatment guidelines, the combination of two drugs should be considered. It allows to achieve the goal of improved efficacy with reduced risk of adverse effects. One of the trials that compared the effectiveness and tolerance of monotherapy with 20 mg rosuvastatin versus drugs combination (ezetimibe 10 mg and rosuvastatin 10 mg) was RACING trial with 3780 participants with atherosclerotic cardiovascular disease. Results have shown that smaller dose of statin combined with ezetimibe was non-inferior to higher-intensity statin monotherapy, but combined therapy allowed to obtain higher percentage of patients with LDL cholesterol concentration of less than 70 mg/dL (respectively after 1, 2 and 3 year) in 73%, 75%, and 72% of them in the combination therapy group (ezetimibe + rosuvastatin) and in 55%, 60%, 58% of participants who were receiving only rosuvastatin in a high dose [45]. It was also confirmed that the combination of ezetimibe with rosuvastatin is capable of reducing LDL-C level even by 61-65% from baseline (20 mg. rosuvastatin; 10 mg. Ezetimibe) [46].
The efficacy of combined treatment strategy has also been demonstrated in patients with homozygous familial hypercholesterolemia. The randomized, double-blind, parallel-group study has shown that ezetimibe with statin in dose of 80 milligrams of atorvastatin or simvastatin significantly reduced LDL-C level compared to statin monotherapy (-27,5% versus -7%) [47]. Davidson and colleagues in a study of 2382 patients have demonstrated that maximal LDL-C drop while using combined therapy, was observed 2 weeks after the initiation of treatment with ezetimibe. Additionally, the safety and supreme potency to decrease LDL-C concentration using coadministration of 10 milligrams of ezetimibe with statin compared to statin alone was noticed [48].
Further safety analyses of combined therapy have also not indicated for a greater propensity to adverse effects compared to statin monotherapy or placebo [49].”
[47] Gagné, C.; Gaudet, D.; Bruckert E. Ezetimibe Study Group. Efficacy and safety of ezetimibe coadministered with atorvastatin or simvastatin in patients with homozygous familial hypercholesterolemia. Circulation. 2002, 105(21):2469-75.
[48] Davidson, M.H.; Ballantyne, C.M.; Kerzner, B.; Melani, L.; Sager, P.T.; Lipka, L.; Strony, J.; Suresh, R.; Veltri, E. Ezetimibe Study Group. Efficacy and safety of ezetimibe coadministered with statins: randomised, placebo-controlled, blinded experience in 2382 patients with primary hypercholesterolemia. Int J Clin Pract. 2004, 58(8):746-55.
[49] Kerzner, B.; Corbelli, J.; Sharp, S.; Lipka, L.J.; Melani, L.; LeBeaut, A.; Suresh, R.; Mukhopadhyay, P.; Veltri, E.P. Ezetimibe Study Group. Efficacy and safety of ezetimibe coadministered with lovastatin in primary hypercholesterolemia. Am J Cardiol. 2003, 91(4):418-24.
Point 3:
In the chapter of PCSK9 inhibitors the author only briefly explained the different forms of PCSK9 and their importance for lipids metabolism. In particular, the plasma presence of a furin-cleaved form and its importance for the regulation of turnover of LDLR should be mentioned together with the appropriate literature.
Response 3:
The paragraph concerning PCSK9 functioning was completed with more detailed description of different forms of PCSK9:
“4.5.1 PCSK9
Plasma LDL-C concentration depends greatly on its removal from the blood by the liver. The effectiveness of this process is related to the expression of LDL receptors (LDLRs) on the hepatocytes’ surface. When LDL attaches to a dedicated receptor, then the formed complex is internalized by clathrin-mediated endocytosis and directed to endosomes. There, LDL detaches from LDLR and is further transported to the lysosomes where undergoes degradation. In turn, the LDLR is transported back to the cell surface and it is able to attach more LDL particles from the systemic circulation [50].
In 2001 in cultured cerebellar granule neurons, scientists identified the presence of neuronal apoptosis regulated convertase-1 (NARC-1). It has also been shown that in these cells NARC-1 mRNA is upregulated in response to their exposure to pro-apoptotic signals. The above-described protein playing a role in the mechanism of apoptosis required for the development of the nervous system is currently known under the name of proprotein convertase subtilisin/kexin 9 (PCSK9) [51, 52].
PCSK9 is one of the most important regulators of the number of LDLRs. It belongs to one of nine known proprotein convertases (PCs) that serve as serine proteases involved in post-translational modifications of propeptides. PCSK9 is expressed mainly in the liver, but its presence in smaller amounts has also been confirmed in the intestines, kidneys and brain [53]. Its gene is located on chromosome 1p32.3 and the protein is synthesized as a 74kDa zymogen in endoplasmic reticulum [53, 54].
PCSK9 consists of a signal peptide, a prodomain, a catalytic domain and a C-terminal domain [50]. It undergoes many posttranslational modifications. One of them is autocatalytic activation by cleavage at the N-terminus which leads to the formation of mature protein destined for being released into the bloodstream [55]
Some of the mature PCSK9 particles undergo additional cleavage by another proprotein convertase called furin [56]. Significant differences between the two mentioned forms of PCSK9 have been shown. It is considered that the mature forms of PCSK9, unlike the furin-cleaved forms, are responsible for regulating the amount of LDLR. Therefore, mature forms seem to relate to atherogenic properties [57, 58, 59]. Furthermore, PCSK9 loss-of-function point mutations that impede recognition and proteolysis by furin enhance LDLR degradation [55]. This regularity has been noted in the group of patients with FH and a mutation of PCSK9 reducing its susceptibility to furin. Increased secretion of the uncleaved protein was associated with higher concentration of LDL-C [58].
After releasing to plasma, PCSK9 binds to epidermal growth factor homology domain A (EGF-A) of the LDL receptor. That leads to endocytosis of a newly formed complex and digestion of LDLR by hepatocyte lysosomes. By reducing the number of LDL receptors on the hepatocyte surface, PCSK9 increases LDL-C level [60, 61]. Some studies have shown that PCSK9 can also lead to LDLR degradation by intracellular pathway, before its exocytosis via Golgi-lysosome route [50, 62, 63].
A breakthrough in understanding the role of PCSK9 was the discovery made by Abifadel and colleagues in 2003 who identified two missense mutations in PCSK9 genes (S127R, F216L) as the cause of autosomal dominant familial hypercholesterolemia [64]. In subsequent years, studies have shown that similar mutations increasing PCSK9 activity (gain-off-function mutations) are associated with increased LDL-C level, while loss-of-function mutations result in decreased LDL-C level and decreased risk of CVD [62].
The discovery of PCSK9 impact on the lipid metabolism paved the way for a novel class of lipid-lowering agents involving PCSK9 inhibition:
- monoclonal antibodies (mAbs) – alirocumab and evolocumab
- small interfering RNA (siRNA) - inclisiran.
Unlike mAbs, which block only the extracellular mechanism of PCSK9 action, inclisiran acts on both intra- and extracellular pathways [50].”
[55] Oleaga C.; Hay J.; Gurcan E.; David LL.; Mueller PA.; Tavori H.; Shapiro MD.; Pamir N.; Fazio S. Insights into the kinetics and dynamics of the furin-cleaved form of PCSK9. J Lipid Res. 2021;62:100003.
[56] Benjannet S.; Rhainds D.; Hamelin J.; Nassoury N.; Seidah NG. The proprotein convertase (PC) PCSK9 is inactivated by furin and/or PC5/6A: functional consequences of natural mutations and post-translational modifications. J Biol Chem. 2006 Oct 13;281(41):30561-72.
[57] Nakamura A.; Kanazawa M.; Kagaya Y.; Kondo M.; Sato K.; Endo H, Nozaki E. Plasma kinetics of mature PCSK9, furin-cleaved PCSK9, and Lp(a) with or without administration of PCSK9 inhibitors in acute myocardial infarction. J Cardiol. 2020 Oct;76(4):395-401.
[58] Kataoka Y.; Harada-Shiba M.; Nakao K.; Nakashima T.; Kawakami S.; Fujino M.; Kanaya T, Nagai T.; Tahara Y, Asaumi Y.; Hori M, Ogura M.; Goto Y, Noguchi T.; Yasuda S. Mature proprotein convertase subtilisin/kexin type 9, coronary atheroma burden, and vessel remodeling in heterozygous familial hypercholesterolemia. J Clin Lipidol. 2017 Mar-Apr;11(2):413-421.e3.
[59] Lipari MT.; Li W.; Moran P.; Kong-Beltran M.; Sai T, Lai J.; Lin SJ.; Kolumam G.; Zavala-Solorio J.; Izrael-Tomasevic A.; Arnott D, Wang J.; Peterson AS, Kirchhofer D. Furin-cleaved proprotein convertase subtilisin/kexin type 9 (PCSK9) is active and modulates low density lipoprotein receptor and serum cholesterol levels. J Biol Chem. 2012 Dec 21;287(52):43482-91.
Point 4
The chapter of anti-PCSK9 monoclonal antibodies is incomplete, missing several references and informations. There are clinical trials not mentioned. Please refer for example to the following similar paper, reporting several studies on the use of PCSK9 Abs:
PMID: 36839781
Response 4:
Authors revised paragraph concerning anti-PCSK9 monoclonal antibodies using suggested reference:
“In the FOURIER study, evolocumab or placebo was administered every two or four weeks to 27,564 patients with ASCVD, LDL-C >70 mg/dL, or non-HDL-C >100 mg/dL who were taking an optimal dosage of statin. The primary combined end point (CV death, myocardial infarction, stroke, hospitalization for unstable angina, or coronary revascularization) occurred at significantly lower rate at 2.2 years of follow-up in the evolocumab group (1344 patients, 9.8%), as compared to the placebo group (1563 patients, 11.3%, p 0,001). Apart from minor increase in incidence of injection site reactions in evolocumab group, there were no significant differences in adverse events between both groups [66, 67].
The ODYSSEY study showed that alirocumab was safe and effective for patients who had experienced an ACS incident in the year prior and were on maximally tolerated statin treatment. A total of 18,924 individuals were recruited and given either with alirocumab 75-150 mg every two weeks (to maintain LDL-C between 25 and 50 mg/dL) or placebo following an initial run-in period of 2-16 weeks on high-intensity statin treatment. MACE were significantly reduced in alirocumab-treated participants compared to placebo group (9,5% vs. 11,1%, p 0,001). [67]. Furthermore, CHOICE II trial has demonstrated alirocumab’s efficacy in statin intolerant patients, which translated into a 56% reduction in LDL (vs placebo; P < 0.001) [68].”
[66] Sabatine, M.S.; Giugliano, R.P.; Keech, A.C.; Honarpour, N.; Wiviott, S.D.; Murphy, S.A.; Kuder, J.F.; Wang, H.; Liu, T.; Wasserman, S.M.; et al. Evolocumab and Clinical Outcomes in Patients with Cardiovascular Disease. N. Engl. J. Med. 2017, 376, 1713–1722.
[67] O’Donoghue, M.L.; Giugliano, R.P.; Wiviott, S.D.; Atar, D.; Keech, A.; Kuder, J.F.; Im, K.; Murphy, S.A.; Flores-Arredondo, J.H.; López, J.A.G.; et al. Long-Term Evolocumab in Patients with Established Atherosclerotic Cardiovascular Disease. Circulation 2022, 146, 1109–1119.
Point 5:
In table 1 all the infos related to LDL-C plasma reduction, PCSK9 reduction, should have the appropriate reference.
Response 5:
Authors added appropriate references:
|
|
Anti-PCSK9 monoclonal antibodies (evolocumab, alirocumab) |
siRNA targeting PCSK9 (inclisiran) |
|
Route of administration |
Injectable (s.c.) |
Injectable (s.c.) |
|
Dose |
140 mg / 420 mg |
284 mg |
|
Dosing frequency |
Every two weeks or once monthly |
0–90–180 days and every 6 months thereafter |
|
LDL-C plasma level reduction 88, 89 |
~ 50-60% |
~ 50% |
|
PCSK9 reduction 82, 89 |
~ 90% |
~ 60-80% |
|
Relative reduction of cardiovascular events |
~15% |
~17% |
|
Mechanism of action |
Blocking of the extracellular interaction of PCSK9 LDLR |
PCSK9 synthesis inhibition through RNA interference |
|
Advantages |
-High specificity |
-No serious adverse events -High specificity -Long term effect -Infrequent dosing |
|
Disadvantages |
-Frequent subcutaneous dosing -Short expiry date -High cost |
-High cost |
|
Adverse effects 90, 91 |
-Injection site reaction -Bronchitis (alirocumab) -Diabetes (evolucumab) -Pancreatitis
|
-Injection site reaction -Bronchitis |
Table 1. Comparison of anti-PCSK9 monoclonal antibodies and inclisiran [82, 88-91]
[82] Ray KK, Troquay RPT, Visseren FLJ, Leiter LA, Scott Wright R, Vikarunnessa S, Talloczy Z, Zang X, Maheux P, Lesogor A, Landmesser U. Long-term efficacy and safety of inclisiran in patients with high cardiovascular risk and elevated LDL cholesterol (ORION-3): results from the 4-year open-label extension of the ORION-1 trial. Lancet Diabetes Endocrinol. 2023 Feb;11(2):109-119. doi: 10.1016/S2213-8587(22)00353-9. Epub 2023 Jan 5. PMID: 36620965.
[88] Coppinger, C.; Movahed, M.R.; Azemawah, V.; Peyton, L.; Gregory, J.; Hashemzadeh M. A Comprehensive Review of PCSK9 Inhibitors. J Cardiovasc Pharmacol Ther. 2022, 27:10742484221100107.
[89] Hermel, M.; Lieberman, M.; Slipczuk, L.; Rana, J.S.; Virani, S.S. Monoclonal Antibodies, Gene Silencing and Gene Editing (CRISPR) Therapies for the Treatment of Hyperlipidemia-The Future Is Here. Pharmaceutics. 2023,15(2):459.
[90] Vijay, N.; Ali, A. Inclisiran: A Game Changer in a Changing Game. J Am Coll Cardiol. 2021, 77 (9) 1194–1196
[91] Gupta, S. LDL Cholesterol, Statins and PCSK 9 Inhibitors. Indian Heart Journal. 2015,67(5):419-424.
Point 6:
Before the conclusion the author could add a paragraph of Future perspectives where they can mention the very promising approach of gene editing using CRISPR/CAS9 technologies as an additional potential therapy against hypercholesterolemia.
Response 6:
Additional paragraph concerning future perspectives for the therapy of hyperlipidemia was added:
“6. Possible future lipid lowering treatment
The promising effects of PCSK9 inhibitors have inspired researchers to look for alternative ways to affect PCSK9-dependent hypolipidemic pathway. An interesting approach involves CRISPR/Cas9 - clustered regularly interspaced short palindromic repeats/CRISPR-associated protein 9 – a mechanism based on genome editing that has been observed for the first time in bacteria as a method of adaptive immunity against viral infections. Fragments of viral DNA derived from previous infections are employed to fight similar bacteriophages attack in the future [103].
CRISPR requires two elements - a guide RNA and Cas proteins. A guide RNA is necessary for sequence-specific targeting, while Cas proteins play structural role. The RNA binds to a Cas protein and cut the target genome. Cas9 protein and a guide RNA binds together and contribute to efficient cleavage and deletions of specific loci, which could render protein-coding genes inactive [104].
The technique of genome editing has been intensively expanded in biomedical research. The usefulness of genome editing techniques has been investigated, inter alia, in the field of cardiovascular disorders [105,106]. Genome-sequencing data gives an opportunity to understand how genetic variations affect lipid disorders. Recent experimental studies suggest that CRISPR gene-editing, which targets PCSK9 might be a promising tool to achieve ultimate resolution of elevated LDL-C due to a lifelong therapeutic effect with just a single dose of drug [103].
According to recent study, the CRISPR/Cas9 method can effectively and permanently damage PCSK9 expression when used in vivo [107]. Ding et al. treated mice with an adenovirus expressing S. pyogenes Cas9 and a guide RNA directed at exon 1 of PCSK9. Significant non-homologous end joining (NHEJ)-mediated mutagenesis was confirmed 3–4 days after injection into mice livers. More importantly, this was accompanied by a substantial drop in blood PCSK9 protein levels by 90% and a concurrent 35–40% reduction in total blood cholesterol levels [108].
Similarly positive effects were obtained in a study using CRISPR/Cas9 in humanized-liver mouse models (Wang et al. [108].) with more than 50% reduction of PCSK9 level [109, 110]. Unlike the previous two researchers, Ran and colleagues used Cas9 derived from S. aureus. It was small enough to be delivered by adeno-associated viral vectors and resulted in a decrease in PCSK9 concentration by >90% and cholesterol by around 40% [108, 110]. VT – 1001 is an open-label phase 1 study to assess the safety of VERVE-101 (CRISPR technology) in a single dose in humans [89].
Apart from the CRISPR strategy, other next-generation PCSK9 inhibitors remain under development:
- Lerodalcibep (LIB003): extracellular PCSK9 - binding fusion protein [110, 111];
- AZD8233 (ION449): antisense oligonucleotide conjugated with GalNAc for intrahepatic PCSK9 degradation [110, 112];
- CIVI-008: capadecursen sodium, third-generation antisense oligonucleotide for PCSK9’s expression inhibition [110];
- MK-0616: oral synthetic cyclic peptide inhibiting PCSK9 for daily administration [110, 114, 115];
- NNC0385-0434: peptide inhibitor of PCSK9 used orally [110];
- CVI-LM001: first-in-class oral small molecule PCSK9 modulator inhibiting PCSK9 transcription and degradation of LDLR mRNA [110, 116];
- PCSK9 vaccine (AT04A, AT06A): mimicking the domain of PCSK9 on its N-terminus in phase I study [111-118].”
[89] Hermel, M.; Lieberman, M.; Slipczuk, L.; Rana, J.S.; Virani, S.S. Monoclonal Antibodies, Gene Silencing and Gene Editing (CRISPR) Therapies for the Treatment of Hyperlipidemia-The Future Is Here. Pharmaceutics. 2023,15(2):459.
[103] Walker, H.E.; Rizzo, M.; Fras, Z.; Jug, B.; Banach, M.; Penson, P.E. CRISPR Gene Editing in Lipid Disorders and Atherosclerosis: Mechanisms and Opportunities. Metabolites. 2021, 9;11(12):857.
[104] Hille F, Charpentier E. CRISPR-Cas: biology, mechanisms and relevance. Philos Trans R Soc Lond B Biol Sci. 2016, 371(1707):20150496.
[105] Chadwick, A.C.; Musunuru, K. Genome editing for the study of cardiovascular diseases. Curr Cardiol Rep. 2017,19(3):22.
[106] Strong, A.; Musunuru, K. Genome editing in cardiovascular diseases. Nat Rev Cardiol. 2017,14(1):11–20.
[107] Ding, Q.; Strong, A.; Patel, K.M.; Ng, S.L.; Gosis, B.S.; Regan, S.N. et al. Permanent alteration of PCSK9 with in vivo CRISPR-Cas9 genome editing. Circ Res. 2014, 15(5):488–92.
[108] Wang, X.; Raghavan ,A.; Chen, T.; Qiao, L.; Zhang, Y.; Ding, Q.; Musunuru, K. CRISPR-Cas9 Targeting of PCSK9 in Human Hepatocytes In Vivo-Brief Report. Arterioscler Thromb Vasc Biol. 2016, 36(5):783-6.
[109] Ran, F. A., L. Cong, W. X. Yan, D. A. Scott, J. S. Gootenberg, A. J. Kriz, B. Zetsche, et al. 2015. “In vivo genome editing using Staphylococcus aureus Cas9.” Nature 520 (7546): 186-191.
[110] Arnold, N.; Koenig W. PCSK9 Inhibitor Wars: How Does Inclisiran Fit in with Current Monoclonal Antibody Inhibitor Therapy? Considerations for Patient Selection. Curr Cardiol Rep. 2022,24(11):1657-1667.
[111] Stein, E.; Toth, P.; Butcher, M.B.; Kereiakes, D.; Magnu, P.; et al. Safety, tolerability and LDL-C reduction with a novel anti-PCSK9 recombinant fusion protein (LIB003): results of a randomized, double-blind, placebo-controlled, phase 2 study. Atherosclerosis.2019;287: e7.
[112] Gennemark, P.; Walter, K.; Clemmensen, N.; et al. An oral antisense oligonucleotide for PCSK9 inhibition. Sci Transl Med 2021;13:eabe9117.
[113] Johns, D.G.; Almonte, A.; Bautmans, A.; Campeau, L.; Cancilla, M.T.; Chapman, J.; et al. The clinical safety, pharmacokinetics, and LDL-cholesterol lowering efficacy of MK-0616, an oral PCSK9 inhibitor. Circulation. 2021,144: e573.
[114] ClinicalTrials.gov NCT05261126. A study of the efficacy and safety of MK-0616 (oral PCSK9 inhibitor) in adults with hypercholesterolemia (MK-0616-008). Available from: https:// clini caltr ials.gov/ ct2/ show/ NCT05 261126. Accessed 17 Jun 2022.
[115] ClinicalTrials.gov NCT05261126. A study of the efficacy and safety of MK-0616 (oral PCSK9 inhibitor) in adults with hypercholesterolemia (MK-0616-008). Available from: https:// clini caltr ials.gov/ ct2/ show/ NCT05 261126. Accessed 17 Jun 2022.
[116] Liu, C.; Chen, J.; Chen, H.; et al. PCSK9 Inhibition: From Current Advances to Evolving Future. Cells. 2022, 11(19):2972.
[118] Sahebkar, A.; Momtazi-Borojeni, A.A.; Banach M. PCSK9 vaccine: so near, yet so far! Eur Heart J. 2021,42(39):4007-4010.

Reviewer 2 Report
Dear Editor,
I carefully read the manuscript "Inclisiran – a revolutionary addition to a cholesterol lowering therapy".
My comments and suggestions for the authors are the following:
- English language needs to be carefully revised and improved. For example, "Simply speaking" is a linguistic expression that does not apply to a scientific article, of course.
- The sub-title "2.1. Assessment of cardiovascular risk" should be more properly placed outside the paragraph "2. Lipid - lowering therapies".
- Line 60: The authors only refer to the SCORE risk charts. However, they should mention also the other available risk charts.
- All the abbreviations should be defined at their first occurrence in the manuscript (e.g. PREDIMED, RACING, ...).
- Most of the information included in the Introduction section of the manuscript is out of context. Do the authors want either to write about all available lipid-lowering therapies or focus on inclisiran? In the first case, maybe the title of the manuscript should be revise.
- Table 1: The authors should also include information regarding the incidence of AEs in treatment with iPCSK9 versus Inclisiran.
- A sub-paragraph 3.2.4 about safety issues should be included in the manuscript.
- In their manuscript, the authors should highly consider to refer also to: doi: 10.1016/j.ahjo.2022.100127, doi: 10.1016/S2213-8587(22)00353-9, doi: 10.1093/eurheartj/ehac594, doi: 10.1080/14740338.2019.1620730 and doi: 10.1007/s11886-022-01782-6.
Author Response
Dear Reviewer 2,
I would like to thank you for careful revision of manuscript entitled “Inclisiran – a revolutionary addition to a cholesterol lowering therapy” and relevant remarks that improve the substantive rank of this work. The replies to your comments are presented as changes in paragraphs with the use of red font.
Point 1:
- English language needs to be carefully revised and improved. For example, "Simply speaking" is a linguistic expression that does not apply to a scientific article, of course.
Response 1:
A thorough language check was performed. We are hoping that at current form it meets Reviewer’s expectations. Alternatively, we may send the manuscript to external editing office.
Point 2:
- The sub-title "2.1. Assessment of cardiovascular risk" should be more properly placed outside the paragraph "2. Lipid - lowering therapies".
Response 2:
According to Reviewer’s suggestion the paragraph “Assessment of cardiovascular risk” was excluded from the paragraph concerning “Lipid lowering therapies” and currently it forms a separate chapter number 3.
Point 3:
- Line 60: The authors only refer to the SCORE risk charts. However, they should mention also the other available risk charts.
Response 3:
Authors edited paragraph concerning the assessment of cardiovascular risk and included data on other risk charts used in clinical practice:
“3. Assessment of cardiovascular risk
Apart from SCORE charts there are also other algorithms that are commonly used to assess the individual’s cardiovascular risk.
The QRISK chart is more often used in the United Kingdom. It was developed by the National Institute for Health and Care Excellence (NICE). In contrast to SCORE2, its unique feature is the inclusion of social deprivation on the assessment of cardiovascular risk [17].
The Framingham risk score has been validated in the United States [18]. The Framingham Heart Study has given the necessary data to develop above-mentioned chart to evaluate the 10-year risk of developing coronary artery disease [19].
The PROCAM system is based on the data obtained from German PROCAM study. 10-year risk of coronary events can be estimated using HDL-C, LDL-C, triglycerides, age, family history of CHD, smoking, diabetes occurrence, and systolic blood pressure [20].”
[17] Hippisley-Cox J, Coupland C, Vinogradova Y, Robson J, Brindle P. Performance of the QRISK cardiovascular risk prediction algorithm in an independent UK sample of patients from general practice: a validation study. Heart. 2008;94(1):34-9.
[18] D'Agostino RB Sr, Grundy S, Sullivan LM, Wilson P; CHD Risk Prediction Group. Validation of the Framingham coronary heart disease prediction scores: results of a multiple ethnic groups investigation. JAMA. 2001;286(2):180-7.
[19] Wilson PW, D'Agostino RB, Levy D, Belanger AM, Silbershatz H, Kannel WB. Prediction of coronary heart disease using risk factor categories. Circulation. 1998, 97(18):1837-47.
[20] Cooper JA, Miller GJ, Humphries SE. A comparison of the PROCAM and Framingham point-scoring systems for estimation of individual risk of coronary heart disease in the Second Northwick Park Heart Study. Atherosclerosis. 2005, 181(1):93-100.”
Point 4:
- All the abbreviations should be defined at their first occurrence in the manuscript (e.g. PREDIMED, RACING, ...).
Response 4:
A thorough text check was performed and all abbreviations were defined at their first occurrence in the manuscript.
Point 5:
- Most of the information included in the Introduction section of the manuscript is out of context. Do the authors want either to write about all available lipid-lowering therapies or focus on inclisiran? In the first case, maybe the title of the manuscript should be revise.
Response 5:
The aim of this manuscript was to review a novel hypolipemic drug from the group of siRNA in context of previous lipid-lowering drugs. Chapter entitled “Introduction” was reshaped according to Reviewer’s suggestion:
“1. Introduction
The definition of dyslipidemia refers to an abnormal concentration of lipids (cholesterol and/or triglycerides and/or lipoproteins) in the blood [1]. Patients with hyperlipidemia are at the higher risk of developing cardiovascular diseases compared to those with normal lipid profile. Statins are the first-line treatment method for lowering LDL-C [2]. However, in some cases the use of statins is insufficient and up to 20% of high-risk patients still cannot reach their LDL-C targets.
In the last several years new drugs targeting proprotein convertase subtilisin/kexin type 9 (PCSK9) have been introduced to the therapy [3]. Inclisiran is a novel long-acting small interfering RNA (siRNA) that inhibits the synthesis of PCSK9 in the liver and thereby contributes to reduction of LDL-C levels. The drug was intended to reduce LDL-C in patients with heterozygous familial hypercholesterolemia or established atherosclerotic cardiovascular disease (ASCVD) who are unable to achieve recommended LDL treatment goals with diet and maximally tolerated statin therapy [4]. Inclisiran is associated with improved lipid profile and according to clinical trials contributes to the reduction of major adverse cardiac events and hospitalizations for heart failure and strokes when compared to placebo. Additionally, the drug has also favorable safety profile with no serious side effects [5].
The aim of this review is to present inclisiran’s mechanism of action, efficacy and safety profile in comparison with currently available lipid lowering drugs.”
Point 6:
- Table 1: The authors should also include information regarding the incidence of AEs in treatment with iPCSK9 versus Inclisiran.
Response 6:
Table 1 was enriched with data on adverse effects during the therapy with anti-PCSK9 mAbs and inclisiran:
|
|
Anti-PCSK9 monoclonal antibodies (evolocumab, alirocumab) |
siRNA targeting PCSK9 (inclisiran) |
|
Route of administration |
Injectable (s.c.) |
Injectable (s.c.) |
|
Dose |
140 mg / 420 mg |
284 mg |
|
Dosing frequency |
Every two weeks or once monthly |
0–90–180 days and every 6 months thereafter |
|
LDL-C plasma level reduction 88, 89 |
~ 50-60% |
~ 50% |
|
PCSK9 reduction 82, 89 |
~ 90% |
~ 60-80% |
|
Relative reduction of cardiovascular events |
~15% |
~17% |
|
Mechanism of action |
Blocking of the extracellular interaction of PCSK9 LDLR |
PCSK9 synthesis inhibition through RNA interference |
|
Advantages |
-High specificity |
-No serious adverse events -High specificity -Long term effect -Infrequent dosing |
|
Disadvantages |
-Frequent subcutaneous dosing -Short expiry date -High cost |
-High cost |
|
Adverse effects 90, 91 |
-Injection site reaction -Bronchitis (alirocumab) -Diabetes (evolucumab) -Pancreatis
|
-Injection site reaction -Bronchitis |
Table 1. Comparison of anti-PCSK9 monoclonal antibodies and inclisiran
[82] Ray KK, Troquay RPT, Visseren FLJ, Leiter LA, Scott Wright R, Vikarunnessa S, Talloczy Z, Zang X, Maheux P, Lesogor A, Landmesser U. Long-term efficacy and safety of inclisiran in patients with high cardiovascular risk and elevated LDL cholesterol (ORION-3): results from the 4-year open-label extension of the ORION-1 trial. Lancet Diabetes Endocrinol. 2023 Feb;11(2):109-119. doi: 10.1016/S2213-8587(22)00353-9. Epub 2023 Jan 5. PMID: 36620965.
[88] Coppinger, C.; Movahed, M.R.; Azemawah, V.; Peyton, L.; Gregory, J.; Hashemzadeh M. A Comprehensive Review of PCSK9 Inhibitors. J Cardiovasc Pharmacol Ther. 2022, 27:10742484221100107.
[89] Hermel, M.; Lieberman, M.; Slipczuk, L.; Rana, J.S.; Virani, S.S. Monoclonal Antibodies, Gene Silencing and Gene Editing (CRISPR) Therapies for the Treatment of Hyperlipidemia-The Future Is Here. Pharmaceutics. 2023,15(2):459.
[90] Vijay, N.; Ali, A. Inclisiran: A Game Changer in a Changing Game. J Am Coll Cardiol. 2021, 77 (9) 1194–1196
[91] Gupta, S. LDL Cholesterol, Statins and PCSK 9 Inhibitors. Indian Heart Journal. 2015,67(5):419-424.
Point 7:
- A sub-paragraph 3.2.4 about safety issues should be included in the manuscript.
Response 7:
A paragraph entitled: “5.2.2 Safety and adverse effects” was edited and enriched with data concerning frequency and severity of side effects during therapy with insliciran:
“5.2.2 Safety and adverse effects
Inclisiran seems to be a solution for people with statin treatment intolerance. ORION trials results showed that it lowers LDL-C levels with a good safety profile. Most of the reported adverse effects of inclisiran were mild and related to injection site reactions like pain or rash. Other off-target effects observed during its use included systemic response resembling infectious symptoms e.g., fever, musculoskeletal pain, sore throat, headache, fatigue or nasopharyngitis [72, 80, 81]. Furthermore, previous trials have shown a similar frequency of side effects in the group of people using placebo and inclisiran - in ORION-1 study 76% in both groups. However, in the group of patients receiving the drug, serious side effects were observed slightly more often - in 11%, compared to 8% in the placebo group [61]. In the cited study one patient experienced an asymptomatic increase in liver enzymes level (γ-glutamyltransferase and alanine aminotransferase), but it was assumed that the adverse event was related to the concomitant use of statin rather than inclisiran itself [61, 72].
In ORION-3 study, a 4-year open-label extension study of ORION-1, side effects were observed in 275 of 284 (97%) patients in inclisiran-only arm and in 80 of 87 (92%) patients in the evolocumab switched to inclisiran arm – mostly mild and self-limiting. In 79 of 284 (28%) in the first group and in 22 of 87 (25%) in the second one, side effects were deemed as related to examined drug. There was a difference in the spectrum of the most common side effects between the groups. The patients in the inclisiran-only arm experienced nasopharyngitis (55 of 284 – 19%), while those in the switching arm suffered from hypertension (17 of 87 – 20%). Serious adverse events occurred 104 of 284 (37 %) of participants in the inclisiran-arm and in 30 of 87 (34%) in the switching arm. When it comes to impaired liver function - there were 10% (28 of 284) of such cases in the inclisiran-only group and 9% (8 of 87) in the switching arm. There were 8 deaths during ORION-3 study – 7 reported in the inclisiran-only group and 1 in the switching group. None of fatal outcomes seemed to be a direct effect of the drug [82].
The safety of inclisiran may be related to its high specificity for hepatocytes. Inclisiran molecules are conjugated to N-acetylgalactosamine carbohydrates (GalNAc), complementary to hepatocyte asialoglycoprotein receptors, which results in rapid uptake only into the liver [60, 61]. Selective action reduces the risk of peripheral neuropathies reported during other siRNA therapies [61, 83]. Phosphorothioate modifications of inclisiran (2'- O - methyl nucleotide and 2' - fluoro nucleotide) increase its stability and reduces the risk of immunogenicity [62]. Nevertheless, these modifications may trigger platelet-activating factor and potentially increase the risk of thrombosis. However, neither this effect has been observed in previous studies, nor the activation of the immune response (both cellular response and the production of pro-inflammatory cytokines such as IL-6 or TNF-α were not affected) [61,80, 84].
The effect of inclisiran on immunogenicity was studied in a group of 1800 patients. The presence of anti-drug antibodies was detected in 1,8% of participants before dosing and in 4,9% during 18 months of drug administration. However, the presence of antibodies has not affected drug efficacy [79]. Comparably, in the group of 4747 patients using alirocumab, the presence of anti-drug antibodies was found in 5,1% compared to 1% in placebo group [85].
It is recommended to avoid the use of inclisiran in the group of pregnant and trying to become pregnant women mainly due to residual content in the liver, which is detectable up to 2 years after one dose [10]. In animal studies, no harmful effects on the well-being of the fetuses have been noted, but its use in pregnant women requires further studies. To date there are no information on the secretion of the drug into the human milk but animal tests have shown this process to occur [10,79].
Inclisiran can be used in patients with impaired renal function and does not require dose adjustment [76]. On the basis of the ORION - 1 and ORION - 7 studies analysis, the pharmacodynamics of the drug were comparable in patients with normal and impaired renal function. Increased plasma concentrations were noted and correlated positively with the degree of renal dysfunction, but the drug was not detectable in plasma in either group at 48h [86, 87].
To date, in contrast to statin therapy, there are no indications that pharmacological inhibition of PCSK9 increases the incidence of diabetes [88].
In patients with hepatic impairment (Child-Pugh class A and B) inclisiran seems to be safe in standard doses. However, it has not been tested in group of patients with severe liver damage (Child-Pugh class C). Therefore, it should be used with caution in this population [79].
Due to the fact that inclisiran does not inhibit or increase cytochrome P450 activity, it is expected that it should not have clinically significant interactions with other drugs, including statins [79]. Main differences between anti-PCSK9 monoclonal antibodies and inclisiran are presented in table 1.”
[82] Ray KK, Troquay RPT, Visseren FLJ, Leiter LA, Scott Wright R, Vikarunnessa S, Talloczy Z, Zang X, Maheux P, Lesogor A, Landmesser U. Long-term efficacy and safety of inclisiran in patients with high cardiovascular risk and elevated LDL cholesterol (ORION-3): results from the 4-year open-label extension of the ORION-1 trial. Lancet Diabetes Endocrinol. 2023 Feb;11(2):109-119.
[88] Coppinger, C.; Movahed, M.R.; Azemawah, V.; Peyton, L.; Gregory, J.; Hashemzadeh M. A Comprehensive Review of PCSK9 Inhibitors. J Cardiovasc Pharmacol Ther. 2022, 27:10742484221100107.
Point 8:
- In their manuscript, the authors should highly consider to refer also to:
doi: 10.1016/j.ahjo.2022.100127,
doi: 10.1016/S2213-8587(22)00353-9,
doi: 10.1093/eurheartj/ehac594,
doi: 10.1080/14740338.2019.1620730 and
doi: 10.1007/s11886-022-01782-6.
Response 8:
Authors implemented data on the safety of therapy with inclisiran in the paragraph 5.2.2.:
“In ORION-3 study, a 4-year open-label extension study of ORION-1, side effects were observed in 275 of 284 (97%) patients in inclisiran-only arm and in 80 of 87 (92%) patients in the evolocumab switched to inclisiran arm – mostly mild and self-limiting. In 79 of 284 (28%) in the first group and in 22 of 87 (25%) in the second one, side effects were deemed as related to examined drug. There was a difference in the spectrum of the most common side effects between the groups. The patients in the inclisiran-only arm experienced nasopharyngitis (55 of 284 – 19%), while those in the switching arm suffered from hypertension (17 of 87 – 20%). Serious adverse events occurred slightly more frequently in the inclisiran-arm only (104 of 284 – 37%) compared to the switching arm (30 of 87 – 34%). When it comes to impaired liver function - there were 10% (28 of 284) of such cases in the inclisiran-only group and 9% (8 of 87) in the switching arm. There were 8 deaths during ORION-3 study – 7 reported in the inclisiran-only group and 1 in the switching group. None of fatal outcomes seemed to be a direct effect of the drug [82].
The safety of inclisiran may be related to its high specificity for hepatocytes. Inclisiran molecules are conjugated to N-acetylgalactosamine carbohydrates (GalNAc), complementary to hepatocyte asialoglycoprotein receptors, which results in rapid uptake only into the liver [60, 61]. Selective action reduces the risk of peripheral neuropathies reported during other siRNA therapies [61, 83]. Phosphorothioate modifications of inclisiran (2'- O - methyl nucleotide and 2' - fluoro nucleotide) increase its stability and reduces the risk of immunogenicity [62]. Nevertheless, these modifications may trigger platelet-activating factor and potentially increase the risk of thrombosis. However, neither this effect has been observed in previous studies, nor the activation of the immune response (cellular response and the production of pro-inflammatory cytokines such as IL-6 or TNF-α were not affected) [61,80, 84].”
”
[82] Ray KK, Troquay RPT, Visseren FLJ, Leiter LA, Scott Wright R, Vikarunnessa S, Talloczy Z, Zang X, Maheux P, Lesogor A, Landmesser U. Long-term efficacy and safety of inclisiran in patients with high cardiovascular risk and elevated LDL cholesterol (ORION-3): results from the 4-year open-label extension of the ORION-1 trial. Lancet Diabetes Endocrinol. 2023 Feb;11(2):109-119. doi: 10.1016/S2213-8587(22)00353-9. Epub 2023 Jan 5. PMID: 36620965.”
[84] Cicero A. F. G.; Fogacci F.; Zambon A.; Toth P. P.; Borghi C.; . Efficacy and safety of inclisiran a newly approved FDA drug: a systematic review and pooled analysis of available clinical studies, American Heart Journal, 2022.
Authors implemented data on future possibilities in lipid-lowering therapies:
“6. Possible future lipid lowering treatment
The promising effects of PCSK9 inhibitors have inspired researchers to look for alternative ways to affect PCSK9-dependent hypolipidemic pathway. An interesting approach involves CRISPR/Cas9 - clustered regularly interspaced short palindromic repeats/CRISPR-associated protein 9 – a mechanism based on genome editing that has been observed for the first time in bacteria as a method of adaptive immunity against viral infections. Fragments of viral DNA derived from previous infections are employed to fight similar bacteriophages attack in the future [103].
CRISPR requires two elements - a guide RNA and Cas proteins. A guide RNA is necessary for sequence-specific targeting, while Cas proteins play structural role. The RNA binds to a Cas protein and cut the target genome. Cas9 protein and a guide RNA binds together and contribute to efficient cleavage and deletions of specific loci, which could render protein-coding genes inactive [104].
The technique of genome editing has been intensively expanded in biomedical research. The usefulness of genome editing techniques has been investigated, inter alia, in the field of cardiovascular disorders [105,106]. Genome-sequencing data gives an opportunity to understand how genetic variations affect lipid disorders. Recent experimental studies suggest that CRISPR gene-editing, which targets PCSK9 might be a promising tool to achieve ultimate resolution of elevated LDL-C due to a lifelong therapeutic effect with just a single dose of drug [103].
According to recent study, the CRISPR/Cas9 method can effectively and permanently damage PCSK9 expression when used in vivo [107]. Ding et al. treated mice with an adenovirus expressing S. pyogenes Cas9 and a guide RNA directed at exon 1 of PCSK9. Significant non-homologous end joining (NHEJ)-mediated mutagenesis was confirmed 3–4 days after injection into mice livers. More importantly, this was accompanied by a substantial drop in blood PCSK9 protein levels by 90% and a concurrent 35–40% reduction in total blood cholesterol levels [108].
Similarly positive effects were obtained in a study using CRISPR/Cas9 in humanized-liver mouse models (Wang et al. [108].) with more than 50% reduction of PCSK9 level [109, 110]. Unlike the previous two researchers, Ran and colleagues used Cas9 derived from S. aureus. It was small enough to be delivered by adeno-associated viral vectors and resulted in a decrease in PCSK9 concentration by >90% and cholesterol by around 40% [108, 110]. VT – 1001 is an open-label phase 1 study to assess the safety of VERVE-101 (CRISPR technology) in a single dose in humans [89].
Apart from the CRISPR strategy, other next-generation PCSK9 inhibitors remain under development:
- Lerodalcibep (LIB003): extracellular PCSK9 - binding fusion protein [110, 111];
- AZD8233 (ION449): antisense oligonucleotide conjugated with GalNAc for intrahepatic PCSK9 degradation [110, 112];
- CIVI-008: capadecursen sodium, third-generation antisense oligonucleotide for PCSK9’s expression inhibition [110];
- MK-0616: oral synthetic cyclic peptide inhibiting PCSK9 for daily administration [110, 114, 115];
- NNC0385-0434: peptide inhibitor of PCSK9 used orally [110];
- CVI-LM001: first-in-class oral small molecule PCSK9 modulator inhibiting PCSK9 transcription and degradation of LDLR mRNA [110, 116];
- PCSK9 vaccine (AT04A, AT06A): mimicking the domain of PCSK9 on its N-terminus in phase I study [111-118].”
[110] Arnold, N.; Koenig W. PCSK9 Inhibitor Wars: How Does Inclisiran Fit in with Current Monoclonal Antibody Inhibitor Therapy? Considerations for Patient Selection. Curr Cardiol Rep. 2022,24(11):1657-1667.”

Round 2
Reviewer 1 Report
The authors addressed my comments making the manuscript more complete.
Reviewer 2 Report
Dear Editor,
I carefully read the revised version of the manuscript that is significantly improved compared to the original version. I warmly recommend its publication in the Journal.